# Validation of a Modified QuEChERS Method for the Determination of Selected Organochlorine Compounds in Honey

**DOI:** 10.3390/molecules28020842

**Published:** 2023-01-14

**Authors:** Radosław Lewiński, Agnieszka Hernik, Monika Liszewska, Brian Buckley, Katarzyna Czaja, Wojciech Korcz, Anna Słomczyńska, Paweł Struciński

**Affiliations:** 1Department of Toxicology and Health Risk Assessment, National Institute of Public Health NIH—National Research Institute, 24 Chocimska, 00-791 Warsaw, Poland; 2Environmental and Occupational Health Sciences Institute, 170 Frelinghuysen Road, Piscataway, NJ 08854, USA

**Keywords:** organochlorine pesticides, persistent organic pollutants, honey, method validation, QuEChERS, GC-µECD

## Abstract

Honey is considered to be a health-promoting food product. Therefore, it is assumed that it should be free of contaminants. Although the use of organochlorine pesticides (OCPs) was banned a few decades ago in developed countries, persistent organic pollutants (POPs) are still detected in various environmental and biological matrices, including food. These contaminants exhibit toxic properties and bioaccumulate in some food chains. The validation of a modified QuEChERS extraction method was successfully performed for *o,p’*-DDT, *o,p’*-DDE, *o,p’*-DDD, *p,p’*-DDT, *p,p’*-DDE, *p,p’*-DDD, heptachlor and dieldrin. 2,2′,4,4′,5,5′-hexachlorobiphenyl (PCB 153) was used as an internal standard. The modification involved changing the solvent from acetonitrile to n-hexane after extraction. Quantitation was carried out using gas chromatography with an electron capture detector (µECD). The mean recovery values for *o,p’*-DDT, *o,p’*-DDE, *o,p’*-DDD, *p,p’*-DDT, *p,p’*-DDE, *p,p’*-DDD and dieldrin, spiked at 2.9 ng/g and 20 ng/g, ranged from 64.7% to 129.3%, and, for heptachlor spiked at 5.6 ng/g and 20 ng/g, ranged from 68.0% to 88.3%. The relative standard deviation (RSD) for these concentrations did not exceed 20%, and the within-laboratory reproducibility was below 20%, except *o,p’*-DDE and *p,p’*-DDT, which were 25.2% and 20.7%, respectively. This modified QuEChERS extraction method for selected organochlorine compounds was demonstrated as effective for routine testing in honey.

## 1. Introduction

Honey is produced by honeybees (*Apis mellifera*) and the main product of beekeeping. Chemically, it is a complex matrix of approximately 300 different compounds. The vast majority are simple sugars (80–83%, depending on its origin) in water. Honey also contains proteins (mainly enzymes), volatile organic compounds, organic acids and numerous vitamins and microelements [1,2]. Due to its health-promoting properties, honey has been used in cosmetic, pharmaceutical and natural medicine products [3,4,5]. It is often seen as a natural panacea and readily consumed by children, the elderly and pregnant women. Honey is also widely used in cooking as a natural sweetener in various dishes.

Due to common knowledge of honey’s beneficial effects on the human body, it is assumed to be free of contaminants. Unfortunately, widespread environmental pollution from various xenobiotics, such as heavy metals, pharmaceuticals or pesticides, is a source of product contamination [6,7,8]. Particular attention is given to organochlorine pesticides (OCPs), persistent organic pollutants (POPs). Since the early 1970s, bans and restrictions on the use of these substances have been introduced in Poland and other European countries; however, even after more than 50 years, they can still be found in the environment [9,10,11,12,13,14]. Wilczynska and Przybylowski reported the presence of dieldrin, *o,p’*-DDT and *p,p’*-DDT in Polish honey at concentrations from traces to 5.93 µg/kg, 18.66 µg/kg and 227.85 µg/kg, respectively [14]. The presence of organochlorine compounds was also reported by Rissato et al. in Portuguese honey (*o,p’*-DDT, *p,p’*-DDT, *p,p’*-DDD and *p,p’*-DDE ranged from 60 µg/kg to 186 µg/kg) and Yavuz et al. in Turkish honey (i.e., heptachlor at 11.6 µg/kg, dieldrin at 3.6 µg/kg or *o,p’*-DDE at 86.2 µg/kg) [15,16]. Testing honey for contaminants with consideration given to the geographic location of the apiary and the restricted bee flight (up to even 10 km from the hive) may be a surrogate for a direct analysis of these contaminants in a given area [17].

In addition to their efficacy as insecticides, organochlorine pesticides have also been shown to have adverse effects on other living residents of the ecosystem, including humans. The high octanol–water partition coefficient of these compounds define their capacity for bioaccumulation and biomagnification. The highest concentrations of these compounds are found in organisms at the upper trophic levels of a food chain, including humans. They are detected in the adipose tissue, e.g., breast adipose tissue, body fluids (e.g., blood) or human milk [18,19,20] and the human brain [21].

The organochlorine compounds may contribute to an increased risk of a number of organ and systemic dysfunctions in humans arising as a consequence of their toxicological properties. They are able to disrupt the endocrine system by interfering with hormones, affecting the synthesis, metabolism and secretion of hormones. Long-term exposure to OCPs have been linked to the development of chronic neurodegenerative diseases, such as Alzheimer’s and Parkinson’s disease, but also to thyroid disease, diabetes and obesity, contributing to a rise in infertility caused by sperm quality impairment or disturbed sex hormone production [22,23,24,25]. The International Agency for Research on Cancer (IARC) has classified DDT and its metabolites into Group 2A as probably carcinogenic to humans [26].

Measured concentrations of the persistent organic pollutants in honey can be used to estimate consumer exposure to these contaminants through ingestion. Young children are likely to be the most exposed group due to their relatively high consumption of honey per body weight [27]. The maximum residue levels (MRLs) set by the European Commission (EC) represent the legal maximum residue amount of a pesticide that can result from the proper use of plant protection products. Exceeding this value prompts a risk assessment linked to the ingestion of pesticides with food (in this case, with honey), but is not necessarily equivalent to a health risk [28]. For the compounds discussed in the paper, the EC has established the following MRLs: for DDT, 0.05 mg/kg; dieldrin, 0.01 mg/kg; heptachlor, 0.01 mg/kg [29]. It should be taken into account that, although organochlorine insecticides are currently considered as environmental contaminants and not active substances of plant protection products, their trace levels in food are treated as pesticide residues under current EU law, and the detection limits reported in this paper meet or exceed those required by the regulation.

Honey is a difficult matrix to analyze because of its high viscosity and sugar content. The complicated matrix requires a more rigorous sample preparation procedure that reduces the analyte response, caused either by a matrix suppression of signal or reduced analyte recovery. To minimize the matrix effect on the analytical results, an internal standard is usually added, and a calibration curve is generated using matrix extracts [30]. Many methods of sample preparation for analysis, including liquid–liquid extraction (LLE), dispersive liquid–liquid microextraction (DLLME) and solid-phase extraction and solid-phase microextraction (SPE and SPME), have previously been reported [15,31,32,33,34]. Despite their advantages, these techniques also have certain limitations. Some are time-consuming (LLE, SPE), whereas other use harmful reagents (DLLME) or require high analysis costs (SPME). QuEChERS provides a cheap, fast and versatile alternative method that is also effective, rugged and safe. It can also be enhanced with additional purification techniques, such as dispersive solid-phase extraction (d-SPE), to improve the results. The most commonly used solid phase for the clean-up of honey extracts is modified silica containing polar primary/secondary amines (PSAs) or non-polar long carbon chains (C18) [35,36,37].

While QuEChERS appears to be the logical choice for the isolation of the POPs from honey, current methods using QuEChERS do not easily compensate for the high-sugar, high-viscosity matrix. The aim of this study was to develop and validate a modified QuEChERS-dSPE method for the determination of DDT, DDD and DDE isomers, as well as heptachlor and dieldrin in honey. The modification of the method required changing the solvent after the extraction from acetonitrile to n-hexane and using an electron capture detector (ECD) for the analysis of selected organochlorine compounds. Using an ECD instead of a mass spectrometry detector (MSD) improves the analytical sensitivity, lower limits of detection and quantitation, while also being more economical. Changing the solvent prevents overloading ECD with acetonitrile, which can overwhelm the analyte signal with background noise. The high polarity and high expansion coefficient of ACN makes it a poor choice to use with GC columns [38].

## 2. Results and Discussion

### 2.1. Linearity

For three compounds (heptachlor, *p,p’*-DDE, *p,p’*-DDD), calibration curves with a determination coefficient R^2^ ≥ 0.999 were obtained, whereas, for *o,p’*-DDD and *o,p’*-DDT, the coefficient R^2^ was >0.998, for the dieldrin and *p,p’*-DDT, the R^2^ was >0.996 and, for *o,p’*-DDE, the R^2^ was >0.993. For the second analyst, curves with R^2^ coefficients of >0.999 were obtained for seven compounds, with only *o,p’*-DDE, R^2^ less than >0.995.

### 2.2. Recovery and Repeatability

Recovery was determined in the fortified honey samples at two concentrations, 2.9 and 20 ng analyte/g of honey, for all of the compounds analyzed. Fortification was carried out in six replicates. The mean recovery values ranged from 66.4% to 79.2% for the higher concentration and from 71.2% to 108.3% for the lower concentration, excluding heptachlor, for which the recovery at the lower concentration was 49.4%. The second analyst obtained slightly better recovery values, ranging from 82.8% to 107.1% for the higher concentrations and from 64.7% to 129.3% for the lower concentration. The mean recovery for heptachlor at 2.9 ng heptachlor/g of honey was 38.8%. The recovery for both analysts for heptachlor at the lower concentration (2.9 ng heptachlor/g of honey) was considered too poor and a 5.6 ng heptachlor/g of honey spike was used as the lower concentration to calculate the recovery. The relative standard deviation (RSD) at both concentrations for all of the compounds was below 20%. Table 1 and Table 2 show selected validation parameters for the analytes measured by both analysts.

The mean recovery and repeatability results were considered acceptable, except as already noted for heptachlor. The results obtained for heptachlor mean recoveries and relative standard deviations of 77.4% and 7.0% for the first analyst and 68.0% and 3.8% for the second analyst, respectively, are shown in Table 3. The lower recovery for the 2.9 ng/g spike was probably caused by the co-elution of matrix constituents.

Figure 1 shows an example of chromatograms of the compounds analyzed in a honey sample fortified at 11.2 ng/g (light gray line) and a non-fortified sample (dark gray line).

### 2.3. Reproducibility

The within-laboratory reproducibility was calculated by averaging the relative standard deviations obtained by the two analysts for all compounds tested. The RSD values at both fortification levels were below 20%, except for *o,p’*-DDE (25.2%) for the lower fortification concentration and *p,p*’-DDT (20.7%) for the higher fortification concentration. The within-laboratory reproducibility for heptachlor, at the fortification level 5.6 ng/g of honey, was 8.7%. Table 4 shows the mean values of the relative standard deviations (RSD) and the mean recovery values from all of the trials for a given compound, representing reproducibility, obtained by both analysts for the compounds tested at the two fortification levels.

According to the SANTE/11312/2021 guide [39] and due to the complex characteristics of the honey matrix (high viscosity, high sugar content, high variability of composition depending on the floral origin), the acceptable mean recovery range was assumed to be from 60% to 130%, with acceptable relative standard deviations and within-laboratory reproducibility of below 20%. At the concentrations tested, the validation parameters obtained were considered acceptable, with exceptions noted.

Table 5 compares the most important validation parameters, i.e., the recovery values and the limit of quantification of compounds analyzed in honey, already reported using different variants of the QuEChERS method, and includes the one applied in this study.

Methods 1 and 2 were carried out using acetonitrile and differed in the detection method. Method 3 involved the complete replacement of the extraction solvent with ethyl acetate. The table shows the limit of quantification (LOQ) values of the methods compared, the detector type and the main extraction solvent.

### 2.4. Role of the Keeper

Due to the initial problems with reproducibility, the effectiveness of *n*-dodecane in the keeper role was tested. For this purpose, two sample sets were prepared (five samples per set) containing 1 mL of honey extracts fortified with 30 ng of each standard. Immediately before solvent evaporation, 70 µL of *n*-dodecane was added to the first set. After solvent evaporation, the residues were reconstituted in 1 mL of *n*-hexane and the standards were quantified. As shown in Table 6, the addition of a keeper (*n*-dodecane) improved the recovery and repeatability for all of the compounds analyzed.

### 2.5. Limit of Quantification

The limit of quantification (LOQ) was considered as the lowest concentration used to generate the calibration curve and set at 2.9 ng/g of honey, except for heptachlor, for which the LOQ was 5.6 ng/g of honey.

## 3. Materials and Methods

### 3.1. Material

The honey used for the validation came from the Masovian Voivodeship. Until the analyses, it was stored at room temperature in a glass vessel.

### 3.2. Reagents

The reagents used in this study include: acetonitrile GC grade (Polskie Odczynniki Chemiczne, POCH, Gliwice, Poland), *n*-hexane pesticide residue analysis grade (POCH, Gliwice, Poland), ready-to-use QuEChERS kits (Agilent, Warsaw, Poland), glacial acetic acid (BDH, Poole, UK) and *n*-dodecane (Merck, Warsaw, Poland). The following certified standards were also: *o,p’*-DDE, *p,p’*-DDE, *o,p’*-DDD, *p,p’*-DDD, *o,p’*-DDT, *p,p’*-DDT, dieldrin and 2,2′,4,4′,5,5′-hexachlorobiphenyl (PCB 153) (Institute of Organic Industry, Warsaw, Poland) and heptachlor (Dr. Ehrenstorfer GmbH). All of the standards had a purity of >99%.

### 3.3. Standard Solutions

Standard solutions for each compound at a concentration of 20 µg/mL were made by preparing appropriate acetonitrile dilutions of individual 200 µg/mL stock solutions. A working solution containing all of the standards at an equal concentration of 1 µg/mL was also made. A PCB 153 was used as the internal standard and prepared separately by as a 20 µg/mL standard solution and then diluting it accordingly. All of the standard solutions were stored in a refrigerator at 4 °C.

### 3.4. Sample Preparation

A modified version of a previously reported QuEChERS method [42] was used in this study. The modification includes introducing an evaporation step and changing the solvent after extraction from acetonitrile to *n*-hexane. A total of 5 g of honey was weighed out and put into a 50 mL tube, to which, 10 mL of water was subsequently added. Then, the tube was shaken manually for 1 min to dissolve the honey. Next, 10 mL of a 1% acetic acid solution in acetonitrile and QuEChERS extraction salts (4 g MgSO_4_, 1 g NaCl, 1 g sodium citrate dihydrate and 0.5 g sodium hydrogencitrate sesquihydrate) were added to the aqueous honey solution, and the tube was shaken vigorously for 1 min. The resulting suspension was centrifuged at 4000 RPM for 3 min, and the resulting supernatant was transferred to a 15 mL tube containing 900 mg MgSO_4_ and 150 mg PSA. The contents of the tube were again vigorously shaken for 1 min and then centrifuged at 4000 RPM for 1 min. From the resulting supernatant, 1 mL was taken and 70 µL of *n*-dodecane and 10 µL of PCB 153 internal standard were added and evaporated to dryness under a light stream of nitrogen. The resulting residue was reconstituted in 1 mL of *n*-hexane and analyzed using GC-ECD. The procedure described above is shown in Figure 2.

### 3.5. Chromatography

The residue analysis was performed using the Agilent 6890N GC (Wilmington, NC, USA) equipped with an Agilent 7683B autosampler (Shanghai, China) and an electron capture detector (ECD) (Wilmington, NC, USA). Separation of the analytes was conducted using an HP-5 ((5%-Phenyl)-methylpolysiloxane) capillary column (30 m × 250 µm id and 0.25 µm film thickness). The following oven temperature program was used: 100 °C (1.7 min)–30 °C min^−1^–210 °C (0 min)–5 °C min^−1^–300 °C (5 min). The injection volume was 5 µL, helium was used as carrier gas at a flow rate of 3.2 mL/min, nitrogen was used as makeup gas at a flow rate of 60 mL/min and the detector temperature was set to 330 °C.

### 3.6. Validation

The following method validation parameters were determined: linearity, recovery, repeatability and within-laboratory reproducibility, as well as limit of quantification. In order to determine the range of linearity, six-point calibration curves were generated, in triplicate per each calibration level, over a concentration range of 2.9–72 ng/g of honey for all of the compounds tested. Due to the strong matrix effect, calibration curves were prepared in the matrix extract by fortifying the extracts with appropriate amounts of the standard solution before changing the solvent. An internal standard (IS) was used to correct for analyte loss during sample preparation, including solvent evaporation. The internal standard signal to its mass ratio is used to adjust the measured analyte response to its corrected concentration, and was used for the further calculations. The blank sample was created from a honey extract spiked with internal standard. The recovery was obtained for honey spiked at two concentrations: 2.9 and 20 ng analyte/g of honey. For heptachlor, validation parameters were measured at 5.6 and 20 ng analyte/g of honey. In order to determine within-laboratory reproducibility, a second analyst repeated the analytical process described above using the same instruments.

## 4. Conclusions

The method for the analysis of selected organochlorine compounds in honey was successfully validated in this paper. The method described was characterized by good recovery and reproducibility values despite the presence of strong matrix effects, as well as simplicity, speed and a relatively low price. The modification, a change in the solvent commonly used in the QuEChERS method from acetonitrile after the extraction step to *n*-hexane, allowed chromatographic techniques coupled with an electron capture detector to be successfully employed.

## Figures and Tables

**Figure 1 molecules-28-00842-f001:**
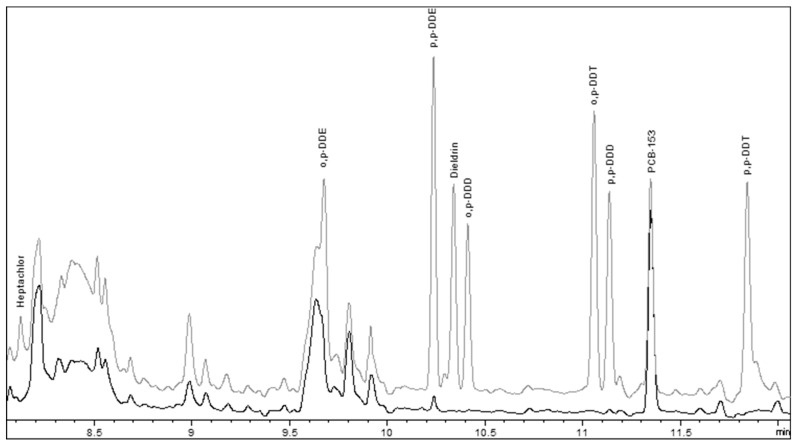
GC-µECD chromatogram of a honey sample fortified with the compounds studied.

**Figure 2 molecules-28-00842-f002:**
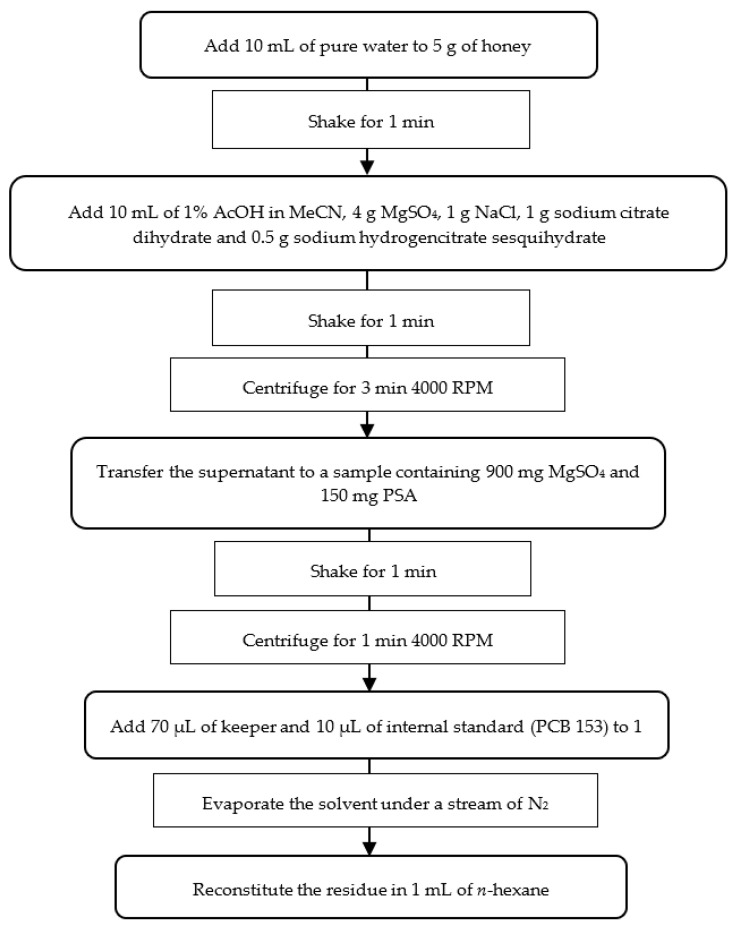
Extraction method used in the study.

**Table 1 molecules-28-00842-t001:** Analyst 1 method validation parameters.

Compound	RT (min)	R^2^	Fortification I (2.9 ng/g)		Fortification II (20.0 ng/g)	
			Mean Recovery (%) n = 6	RSD (%)	Mean Recovery (%) n = 6	RSD (%)
Heptachlor	8.126	0.9990	49.4	1.9	69.6	5.6
*o,p’*-DDE	9.663	0.9939	82.9	15.0	73.7	3.2
*p,p’*-DDE	10.236	0.9994	81.7	4.0	67.0	3.2
Dieldrin	10.336	0.9967	108.3	9.3	79.2	4.7
*o,p’*-DDD	10.410	0.9986	81.7	2.5	72.9	2.6
*o,p’*-DDT	11.057	0.9988	78.0	3.2	71.2	2.7
*p,p’*-DDD	11.135	0.9990	71.2	3.8	68.0	4.2
*p,p’*-DDT	11.841	0.9968	92.1	8.2	66.4	6.8

RT—retention time, R^2^—coefficient of determination, RSD—relative standard deviation, n—number of replicates.

**Table 2 molecules-28-00842-t002:** Analyst 2 method validation parameters.

Compound	RT (min)	R^2^	Fortification I (2.9 ng/g)		Fortification II (20.0 ng/g)	
			Mean Recovery (%) n = 6	RSD (%)	Mean Recovery (%) n = 6	RSD (%)
Heptachlor	8.124	0.9992	38.8	14.6	88.3	10.0
*o,p’*-DDE	9.654	0.9956	129.3	11.0	104.3	5.4
*p,p’*-DDE	10.224	0.9996	64.7	4.8	82.8	4.4
Dieldrin	10.326	0.9997	92.3	18.0	107.1	6.0
*o,p’*-DDD	10.399	0.9996	75.0	6.0	95.2	2.2
*o,p’*-DDT	11.043	0.9999	70.6	4.3	92.1	2.6
*p,p’*-DDD	11.122	0.9999	73.7	5.7	89.8	3.1
*p,p’*-DDT	11.827	0.9999	95.5	9.8	97.4	8.6

RT—retention time, R^2^—coefficient of determination, RSD—relative standard deviation, n—number of replicates.

**Table 3 molecules-28-00842-t003:** Validation of heptachlor at 5.6 ng/g.

Compound	Analyst I				Analyst II			
	RT (min)	R^2^	Mean Recovery (%) n = 6	RSD (%)	RT (min)	R^2^	Mean Recovery (%) n = 6	RSD (%)
Heptachlor	8.105	0.9994	77.4	7.0	8.105	0.9985	68.0%	3.8

RT—retention time, R^2^—coefficient of determination, RSD—relative standard deviation, n—number of replicates.

**Table 4 molecules-28-00842-t004:** Method reproducibility.

Compound		Heptachlor	*o,p’*-DDE	*p,p’*-DDE	Dieldrin	*o,p’*-DDD	*o,p’*-DDT	*p,p’*-DDD	*p,p’*-DDT
Fortification I (2.9 ng/g)	Mean recovery (%), n = 12	44.1	106.1	73.2	100.3	78.3	74.3	72.5	93.8
	RSD (%)	15.1	25.2	12.4	15.8	6.2	6.2	5.2	9.3
Fortification II (20 ng/g)	Mean recovery (%), n = 12	79.0	89.0	74.9	93.2	84.0	81.7	78.9	81.9
	RSD (%)	14.7	17.8	11.3	16	13.4	13.1	14.3	20.7

RSD—relative standard deviation, n—number of replicates.

**Table 5 molecules-28-00842-t005:** Method comparison.

Method	Compound	LOQ (ng/g)	Recovery (%) ^1^	Detection	Solvent	References
Method 1	*o,p’*-DDD	3.7	89.0 (10)	GC-MS	Acetonitrile	[36]
	*p,p’*-DDT	65.9	91.0 (60)			
	Dieldrin	29.5	90.0 (30)			
Method 2	*p,p’*-DDE	10	76.0 (20)	GC-ECD	Acetonitrile	[40]
	Dieldrin	10	76.0 (20)			
	Heptachlor	10	71.0 (20)			
Method 3	Heptachlor	33	128.0 (70)	GC-ECD	Ethyl acetate	[41]
	Dieldrin	8	91.6 (70)			
Method presented	*p,p’*-DDE	2.9	81.7 (2.9)	GC-ECD	Acetonitrile	[–]
	*o,p’*-DDD	2.9	81.7 (2.9)			
	*p,p’*-DDT	2.9	92.1 (2.9)			
	Dieldrin	2.9	108.3 (2.9)			
	Heptachlor	5.6	77.4 (5.6)			

LOQ—limit of quantification; ^1^ In brackets, the concentration value [ng/g] for which the recovery was determined.

**Table 6 molecules-28-00842-t006:** The effect of n-dodecane addition on recovery and repeatability.

Compound	Concentration (ng/mL)	Determination without *n*-Dodecane (n = 5)			Determination with *n*-Dodecane (n = 5)		
		Concentration Measured(ng/mL)	Recovery (%)	RSD (%)	Concentration Measured(ng/mL)	Recovery (%)	RSD (%)
Heptachlor	30.0	19.8	65.9	22.7	35.9	119.9	4.6
*o,p’*-DDE	30.4	14.5	47.7	24.3	31.5	103.7	3.7
*p,p’*-DDE	30.0	20.5	68.2	11.6	32.7	109.1	2.0
Dieldrin	30.5	13.5	44.4	27.8	34.2	112.3	3.9
*o,p’*-DDD	30.3	15.9	52.7	18.9	31.9	105.3	2.5
*o,p’*-DDT	30.4	15.7	51.6	17.2	30.6	100.7	1.4
*p,p’*-DDD	30.2	18.0	59.6	12.8	31.5	104.2	1.9
*p,p’*-DDT	30.1	15.9	52.7	12.3	27.8	92.2	5.5

RSD—relative standard deviation, n—number of replicates.

## Data Availability

All experimental data are provided in the article.

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
