# Peer review of "Validation of a Modified QuEChERS Method for the Determination of Selected Organochlorine Compounds in Honey"

_molecules, 2023, doi:10.3390/molecules28020842_

Round 1

Reviewer 1 Report

The manuscript deals with validation of a well-established method for the determination of organochlorine compounds in honey samples. The analytes selected were banned quite a while ago and only the validation parameters have been presented. No information was given for their actual levels if any encountered in real samples. The only modification of the QuEChERS method is the change of the solvent and no optimization was made. The manuscript lacks of originality and results for the sample application. The conflict between the recovery values of heptachlor in the text and the table is not clear. 

Author Response

Dear Reviewer,

Thank you very much for your comments and suggestions on how to improve our manuscript. In response to your kind review I would like to address the points that have been raised. 

As suggested by the reviewer, information on the current levels of organochlorine compounds in honey has been added to the manuscript with appropriate citations (see lines 46-51).

Referring to the comments regarding the lack of novelty and no optimization in the method presented, we want to point out that while not quite a new approach to QuEChERS we improved evaporation step (i.e. by using keeper) making it more reproducible and successfully validated this method at very low concentrations. Honey is a difficult matrix and insufficient purification of the extracts after the QuEChERS procedure, is one of the main disadvantages in all QuEChERS methods. We believe our method significantly improved this step. Moreover, we believe this is an alternative approach for any laboratory trying to quantify organochlorine compounds in honey samples but who cannot afford mass spectrometric detection.

Recovery values of heptachlor presented in text fully correspond to values presented in tables. We clarified sentences which were misleading in our manuscript related to heptachlor’s recovery (see lines 116-117).

With kind regards,

Radosław Lewiński.

Reviewer 2 Report

The authors propose the validation of a method for the analysis of 8 organochlorine pesticides in honey. The extraction of the compounds is performed by extraction with modified QuEChERS. Quantification was carried out using gas chromatography with electron capture detector (µECD). This modification consists of drying the extract obtained from the protocol to dryness and reconstituting it with hexane, in order to prevent overloading ECD with acetonitrile.

The paper does not present any novelty. The authors emphasize the novelty of eliminating the solvent, in this case acetonitrile, by another one compatible with gas chromatography, such as hexane. This is not new in a method since many authors perform this same technique to be able to work with both liquid and gas chromatography (see bibliography below)

On the other hand, the detector µECD used by the authors, although it is cheaper than a mass detector, is not more sensitive, since it presents the problem of coelucion with components of the honey matrix. Moreover, it is not an adequate technique, taking into account in recent decades it is sought to find the largest amount of compounds simultaneously (100-600 components), multiresidue methods with maximum sensitivity. And this is only achieved by means of a mass spectrometry detector.

For all the above mentioned, I consider that the article has no scientific relevance and should be rejected for publication.

Lasheras, R.J.; Lázaro, R.; Burillo, J.C.; Bayarri, S. Occurrence of Pesticide Residues in Spanish Honey Measured by QuEChERS Method Followed by Liquid and Gas Chromatography–Tandem Mass Spectrometry. Foods 202110, 2262. https://doi.org/10.3390/foods10102262

Ly, T. K., Ho, T. D., Behra, P., & Nhu-Trang, T. T. (2020). Determination of 400 pesticide residues in green tea leaves by UPLC-MS/MS and GC-MS/MS combined with QuEChERS extraction and mixed-mode SPE clean-up method. Food chemistry326, 126928.

GaweÅ‚, M., Kiljanek, T., Niewiadowska, A., Semeniuk, S., Goliszek, M., Burek, O., & Posyniak, A. (2019). Determination of neonicotinoids and 199 other pesticide residues in honey by liquid and gas chromatography coupled with tandem mass spectrometry. Food Chemistry282, 36-47.

Gil García, M. D., Martínez Galera, M., Uclés, S., Lozano, A., & Fernández-Alba, A. R. (2018). Ultrasound-assisted extraction based on QuEChERS of pesticide residues in honeybees and determination by LC-MS/MS and GC-MS/MS. Analytical and bioanalytical chemistry410(21), 5195-5210.

Shendy, A. H., Al-Ghobashy, M. A., Mohammed, M. N., Alla, S. A. G., & Lotfy, H. M. (2016). Simultaneous determination of 200 pesticide residues in honey using gas chromatography–tandem mass spectrometry in conjunction with streamlined quantification approach. Journal of Chromatography A1427, 142-160.

Li, Y., Kelley, R. A., Anderson, T. D., & Lydy, M. J. (2015). Development and comparison of two multi-residue methods for the analysis of select pesticides in honey bees, pollen, and wax by gas chromatography–quadrupole mass spectrometry. Talanta140, 81-87.

Nadaf, H. A., Yadav, G. S., & Kumari, B. (2015). Validation and monitoring of pesticide residues in honey using QuEChERS and gas chromatographic analysis. Journal of Apicultural Research54(3), 260-266.

Author Response

Dear Reviewer,

Thank you very much for your comments and suggestions on how to improve our manuscript. In response to your kind review I would like to discuss the points you raised in more detail. 

Referring to the comment regarding the lack of novelty we wish to point out that while not quite a new approach to QuEChERS, we improved evaporation step (i.e. by using a keeper) resulting in a more reproducible method and we successfully validated this method at very low concentrations. Honey is a difficult matrix and insufficient purification of the extracts after the QuEChERS procedure, is one of the main disadvantages in all QuEChERS methods. We believe our method significantly improved this step. Moreover, we believe this is an alternative approach for any laboratory trying to quantify organochlorine compounds in honey samples for who cannot afford mass spectrometric detection.  We believe it is very important to present new methods or method modifications that provide alternatives to mass spectrometric detection as the number of labs throughout the world that just cannot afford those instruments, are more than most realize.

The analytes used in this study were chosen to demonstrate that compounds analogous to this ones tested could be quantified in honey with routine methods, using an ECD detector. To the best of our knowledge the ECD detector quantifies compounds containing highly electronegative moieties, at a lower level than single quadrupole MSDs. We recognize that within limits, an MSD is often able to identify as well as quantify analytes but this is not always true and we believe there is always room for analytical methods with fewer capabilities if there is significant cost savings that accompany those savings.

With kind regards,

Radosław Lewiński.

Round 2

Reviewer 1 Report

All the necessary changes have been made.

Author Response

Dear Reviewer,

Thank you very much for your previous comments and suggestions, I sincerely appreciate all your guidance on how to improve our manuscript.

With kind regards,

Radosław Lewiński.

Reviewer 2 Report

There are many works that also use a preservative to improve analyte recovery; this is not new from the authors. It is true that there are laboratories that cannot afford mass detection equipment due to its high cost, among others. Nevertheless, the question is to find out if a honey has or not certain unwanted polluting compounds, such as pesticides. With this technique, the necessary level of sensitivity is not reached to be able to affirm if a sample has an absence of a certain pesticide, a fact that is aggravated if we talk about organic honey. The sensitivity required today by honey packers and marketers is more demanding, reaching levels between 0.01-0.001 mg/Kg. Very far from the technique proposed by the authors.

There are many works that also use a preservative to improve analyte recovery; this is not new from the authors. It is true that there are laboratories that cannot afford mass detection equipment due to its high cost, among others. Nevertheless, the question is to find out if a honey has or not certain unwanted polluting compounds, such as pesticides. With this technique, the necessary level of sensitivity is not reached to be able to affirm if a sample has an absence of a certain pesticide, a fact that is aggravated if we talk about organic honey. The sensitivity required today by honey packers and marketers is more demanding, reaching levels between 0.01-0.001 mg/Kg. Very far from the technique proposed by the authors.

Author Response

Dear Reviewer,

I sincerely appreciate your contribution on how to improve our manuscript. In response to your kind review I would like to discuss the points you raised in more detail. 

With reference to usage of preservative, we recognize the evaporation step as a bottleneck of this method, which has the greatest impact on validation parameters. Perhaps the addition of the keeper to QUeChERS methods is not especially novel but we believe it is necessary to keep recovery values and relative standard deviations at acceptable levels. As we describe, this step significantly improves the reproducibility of this method.

Referring to Maximum Residue Levels (MRLs) established by European Food Safety Authority (EFSA) for these compounds in honey (MRLs respectively for ΣDDT 0,05 mg/kg; dieldrin 0,01 mg/kg; heptachlor 0,01 mg/kg), we believe that LOQs described in our paper meets requirements for detection and determination of pesticides residues at sufficient levels, especially for risk assessment associated with the ingestion of these substances in honey. Nevertheless, we wish to mention that we have future plans to further develop this method and carry out the determination in environmental samples and to also perform a risk assessment.

We have added additional information regarding MRLs in the manuscript (see lines 75-85 and 344-348).

With kind regards,

Radosław Lewiński.